# Programmed Cell Death Ligand 1 Immunohistochemical Expression and Cutaneous Melanoma: A Controversial Relationship

**DOI:** 10.3390/ijms25010676

**Published:** 2024-01-04

**Authors:** Vincenzo Fiorentino, Cristina Pizzimenti, Mariausilia Franchina, Ludovica Pepe, Fernanda Russotto, Pietro Tralongo, Marina Gloria Micali, Gaetano Basilio Militi, Maria Lentini

**Affiliations:** 1Department of Human Pathology in Adult and Developmental Age “Gaetano Barresi”, University of Messina, 98125 Messina, Italy; mariausilia.franchina@studenti.unime.it (M.F.); ludopepe97@gmail.com (L.P.); russottofernanda@gmail.com (F.R.); micalimarina@yahoo.it (M.G.M.); maria.lentini@unime.it (M.L.); 2Department of Biomedical, Dental, Morphological and Functional Imaging Sciences, University of Messina, 98125 Messina, Italy; cristina.pizzimenti@unime.it; 3Department of Women, Children and Public Health Sciences, Catholic University of the Sacred Heart, Agostino Gemelli IRCCS University Hospital Foundation, 00168 Rome, Italy; pietrotralongo@gmail.com; 4Department of Sciences for Promotion of Health and Mother and Child Care, Anatomic Pathology, University of Palermo, 90133 Palermo, Italy; gaetanobmiliti@gmail.com

**Keywords:** cutaneous melanoma, immune surveillance, PD-1/PD-L1 axis, immunotherapy

## Abstract

Cutaneous melanoma (CM) is traditionally considered one of the most “immunogenic” tumors, eliciting a high immune response. However, despite the presence of tumor-infiltrating lymphocytes (TILs), melanoma cells use strategies to suppress antitumor immunity and avoid being eliminated by immune surveillance. The PD-1 (programmed death-1)/PD-L1 (programmed death-ligand 1) axis is a well-known immune escape system adopted by neoplastic cells. Therefore, immunotherapy with PD-1 and PD-L1 inhibitors is quickly becoming the main treatment approach for metastatic melanoma patients. However, the clinical utility of PD-L1 expression assessment in CM is controversial, and the interpretation of PD-L1 scores in clinical practice is still a matter of debate. Nonetheless, the recent literature data show that by adopting specific PD-L1 assessment methods in melanoma samples, a correlation between the expression of such a biomarker and a positive response to PD-1-based immunotherapy can be seen. Our review aims to describe the state-of-the-art knowledge regarding the prognostic and predictive role of PD-L1 expression in CM while also referring to possible biological explanations for the variability in its expressions and related treatment responses.

## 1. Introduction

Cutaneous melanoma (CM) is a highly aggressive type of skin cancer with a high mortality rate; its incidence has been increasing over the last few decades, making it a major public health concern [1]. In fact, with an estimated 97,610 new cases in 2023 in the United States, melanoma of the skin represents 5% of all new cancer cases, with 7990 estimated deaths (1.3% of all cancer deaths) [2]. The incidence varies across different areas of the world, with greater incidence rates observed in regions with higher levels of ultraviolet (UV) radiation exposure, such as Australia, New Zealand, and parts of Europe [3]. Other risk factors for cutaneous melanoma are represented by fair skin, a history of sunburns, a family history of melanoma, and certain genetic mutations [4].

In fact, all melanoma types are characterized by recurrent genetic alterations, and driving mutations tend to be in signaling pathways that regulate proliferation (*BRAF*, *NRAS*, and *NF1*), growth and metabolism (*PTEN* and *KIT*), resistance to apoptosis (*TP53*), duration of the cell cycle (*TERT*), cell identity (*ARID2*), and cell cycle control (*CDKN2A*) [5]. More than two-thirds of melanomas contain mutations in the MAP kinase (MAPK) signaling pathway, which is involved in controlling proliferation and survival in response to growth factors, so when mutations cause its constitutive activity, the consequence is uncontrolled cell growth. About 50% of melanomas contain activating *BRAF* mutations, the most common of which is the V600E mutation, which leads to the constitutive activity of downstream MAPK signaling. Another 15–20% harbor *NRAS* mutations, 2% *CKIT* mutations (common in mucosal melanomas), and 50% of uveal melanomas have *GNAQ* mutations [5,6,7,8]. In particular, cutaneous melanoma is characterized by a complex genetic landscape, and large-scale genomic studies have identified several genetic alterations and mutations associated with its development and progression, particularly regarding the *BRAF*, *NRAS*, and *KIT* genes [4,9].

At present, diagnosis, prognosis, and treatment rely upon the American Joint Committee on Cancer (AJCC) classification of melanoma (eighth edition), which considers several clinicopathological risk factors such as tumor thickness, ulceration, mitotic rate, sentinel lymph node status, and locoregional and distant metastases, in order to stratify neoplasms in four stages [from 0 (in situ melanoma) to IV (metastatic melanoma)] [10,11].

Therefore, staging provides valuable information for treatment planning, prognostic estimation, and the stratification of patients in clinical trials.

However, the AJCC classification system lacks the ability to accurately predict the unique progression and diverse treatment responses of CMs classified at the same stage. In fact, the outcome is influenced not only by the genetic landscape but also by a variety of intricate interactions between the tumor and the host’s immune system [12,13].

## 2. The Role of the Immune System in Cutaneous Melanoma

CM has traditionally been regarded as one of the most “immunogenic” tumors; in fact, it elicits a high immune response in most cases: T-lymphocytes, dendritic cells, macrophages, neutrophils, mast cells, and B-lymphocytes are typically found in CM microenvironment [14] and are involved in anti-melanoma cytotoxicity [15]. Furthermore, the presence of dense tumor-infiltrating lymphocytes (TILs) has been linked to a favorable prognosis [12,13,16], and already Clark et al. in 1989 showed that TILs could represent an independent survival predictor in CM [17]. Moreover, it has been demonstrated that TIL grade can be seen as an independent predictive factor for sentinel lymph node status in CM [12]. Hussein et al. [18] found a progressive rise in TILs during melanogenesis, which was thought to result from rising tumor antigenicity. Interestingly, they observed a reduction in TILs in metastatic melanomas compared to primary tumors, suggesting an immune system failure eventually followed by the evasion of cancer cells from immune surveillance.

## 3. Mechanisms of Immune System Evasion in Cutaneous Melanoma

A significant contradiction in tumor immunology is the progression of melanoma despite the presence of TILs. Unfortunately, melanoma cells use a variety of strategies to suppress antitumor immunity and avoid being eliminated via immune surveillance (Figure 1).

First, the immune system becomes functionally exhausted after prolonged exposure to melanoma antigens. This process consists of overactivation inhibitory checkpoints on immune cells and a negative feedback loop for cytotoxic T-cells [19]. Parallel to this, an additional mechanism that supports the poor cytotoxicity of T-cells is the enrichment of the melanoma environment in tumor-associated macrophages (TAMs), regulatory T cells (T reg), and myeloid-derived suppressor cells (MDSCs). The overproduction of immunosuppressive cytokines (IL-10, TGFβ1, and TGFβ2) and the enzyme indoleamine 2,3-dioxygenase (IDO), as well as the loss of both class I and class II antigens of the major histocompatibility complex (MHC), are the primary causes of the ineffective killing of malignant cells [19].

However, the most famous immunity escape system adopted by neoplastic cells is based on the PD-1 (programmed death-1)/PD-L1 (programmed death-ligand 1) axis. PD-1 was first identified in 1992 as a putative mediator of apoptosis, and subsequently, its role in modulating the hyperstimulated immune system was also discovered [20].

PD-1 is a type I transmembrane protein that belongs to the B7/CD28 family of receptors expressed in humans on T-lymphocytes. After TCR (T cell receptor) stimulation, PD-1 binds to PD-L1 (also known as B7H1) and PD-L2 (programmed death-ligand 2, also known as B7DC), which are constitutively present on antigen-presenting cells (APCs) or in non-hematopoietic tissues upon stimulation of pro-inflammatory cytokines, thus forming the “PD-L1/PD-1 axis”. The PD-L1/PD-1 system regulates immune responses mainly through the mechanism of intracellular inhibitory signaling in effector T cells and regulatory T cells. PD-L1 plays a key role in the immune escape of cancer cells.

The structure of PD-1 consists of an immunoglobulin variable region (IgV), a transmembrane region immunoreceptor (ITIM), and an immunoreceptor (ITSM) [21].

The interaction between PD-1 and PD-L1 induces ITIM and ITSM phosphorylation of the intracellular domain of PD-1, which recruits the acid tyrosine phosphatases Src homolog phosphatase 1 (SHP-1) and Src homolog phosphatase 2 (SHP-2). These phosphatases dephosphorylate several key proteins in T cell antigen activation, hindering T cell cycle progression and related protein expression and ultimately inhibiting cytokine production and T cell proliferation and differentiation, causing loss of immune system function [22].

The PD-L1/PD-1 axis is essential for controlling the continuous activation and proliferation of different immune effectors; when PD-1 engages its ligands, it thus induces a state of T-lymphocyte dysfunction called T-lymphocyte exhaustion [23]. In addition to regulating conventional T-lymphocytes, PD-L1 on APCs can control regulatory T-cell differentiation and immunosuppressive activity. PD-L1 expression may be either constitutive when expressed at a low level, for example, on resting lymphocytes or APCs, or inducible when its expression is upregulated by specific stimuli, such as an inflammatory event [24,25,26]. Nonetheless, a complementary role to PD-L1 function in regulating T-cell activity is exploited by cytotoxic T-lymphocyte-associated protein 4 (CTLA-4), a critical immune checkpoint receptor that exerts regulatory effects on T-lymphocyte activation, is constitutively expressed in regulatory T cells, and is upregulated in conventional T cells subsequent to activation [23].

Notably, under normal conditions, the immune system has an immune surveillance function; when malignant cells appear, the immune system can recognize and specifically remove them as “non-self” cells, inhibiting tumor growth. However, in some cases, malignant cells prohibit immune responses against tumors, resulting in a mechanism of immune escape and tumor immortalization [27] based on the PD-1/PD-L1 signaling pathway. By virtue of the above, recent anticancer treatments have been developed that successfully reactivate the immune response against cancer cells by blocking the interactions between PD-1 and PD-L1 and thus restoring the function of T-lymphocytes. Although TILs and the expression of the PD-L1 protein are frequently associated with melanomas, their influence on prognosis is still debatable [28].

## 4. PD-L1 Expression in Cutaneous Melanoma

The assessment of PD-L1 expression in CM has emerged as an important part of understanding the tumor immune milieu and predicting therapeutic response.

PD-L1 expression can be influenced by various factors (Table 1), including melanoma subtype, mutation burden, and immune-related gene expression [29,30]. For example, uveal melanomas have been reported to have lower PD-L1 expression compared to CMs, while desmoplastic melanomas have shown higher PD-L1 expression [31].

Furthermore, PD-L1-negative tumors have been found to have a lower mutation burden and differential expression of immune-related genes, which may contribute to worse survival outcomes in stage III melanoma [32]. For patients with metastatic melanoma, PD-1 and PD-L1 inhibitors, in particular, are quickly becoming the main treatment approach. Although there is a correlation between PD-L1 expression and response to anti-PD-1/anti-PD-L1 immunotherapy, the thresholds for response depend greatly on the tumor histology and immunohistochemical technique. In fact, in CM, PD-L1 is commonly assessed using immunohistochemical assays, in which tumor tissue samples are stained with specific antibodies directed against such an antigen. Several scoring systems have been employed, including the tumor proportion score (TPS) [33], the combined positive score (CPS) [33,34], and the melanoma (MEL) score [33,35] (Table 2). The TPS measures the percentage of tumor cells showing PD-L1 expression, while the CPS considers both tumor cells and immune cells expressing PD-L1. The MEL score combines the TPS and CPS to provide a comprehensive assessment of PD-L1 expression: it is based on the Allred score, a scoring method used for hormone receptor evaluation in breast cancer [36].

Specifically, the MEL score focuses on the proportion of cells that exhibit positive staining, concerning both tumor cells and mononuclear inflammatory cells located within or close to neoplastic elements, excluding scattered stromal inflammatory cells distant from neoplastic elements. Within a hierarchical scale consisting of six tiers ranging from 0 to 5, scores falling between 2 and 5 are considered positive, indicating a staining level of at least 1%. Conversely, scores of 0 or 1 are classified as negative, indicating a staining level below 1%. The significance of different scoring systems, different PD-L1 antibody clones, and the optimal “cut-off” level remains uncertain and represents a matter of debate in clinical practice.

In the literature, many clones have been used in diverse ways. The 28-8 (Dako) has been employed in the CheckMate067 study, a randomized, double-blind, phase 3 trial involving patients with previously untreated advanced melanoma [37,38]. Patients were randomly assigned to receive either nivolumab (anti-PD-1 antibody) alone, ipilimumab (anti-CTLA4 antibody) alone, or nivolumab plus ipilimumab. The primary endpoint was progression-free survival, and secondary endpoints included overall survival, objective response rate, and safety. A total of 1296 patients were enrolled, and 945 underwent randomization, with 314 patients in the nivolumab-plus-ipilimumab group, 316 in the nivolumab group, and 315 in the ipilimumab cohort. Of the enrolled patients, 610 were male, and 335 were female. The mean age was 59.6 years, ranging from 18 to 90 years. The study was conducted from July 2013 through March 2014, and the minimum follow-up from the date on which the last patient underwent randomization was 60 months. The study did not report any significant differences in outcomes based on age or sex, but a sustained long-term overall survival at 5 years was observed in a greater percentage of patients who received nivolumab plus ipilimumab or nivolumab alone than in those who received ipilimumab alone. Notably, the combination of nivolumab and ipilimumab demonstrated favorable overall survival outcomes in patients with both normal and elevated lactate dehydrogenase levels. Interestingly, tumor PD-L1 expression alone was not predictive of efficacy outcomes, and no significant relationship between PD-L1 expression and improvement in progression-free survival (PFS) in patients receiving combined immunotherapy was found. Moreover, combined therapy led to better objective response rates, progression-free survival, and overall survival than ipilimumab alone, regardless of PD-L1 expression. Finally, adverse events were reported in all three treatment groups, with the overall incidence of grade 3 or 4 drug-related adverse events being higher in the combination group as compared with ipilimumab alone.

A different PD-L1 testing antibody was used in the Keynote006 trial [39], which employed the 22C3 clone (Dako) and compared the efficacy and safety of pembrolizumab (an anti-PD-1 antibody) and ipilimumab in patients with advanced melanoma. It was an open-label, multicenter, randomized, controlled, phase 3 trial conducted in patients aged 18 years or older, with an Eastern Cooperative Oncology Group performance status of 0 or 1, and who had received up to one previous systemic therapy for advanced disease with a known *BRAF* V600 status. The study involved a total of 834 participants who were randomly assigned in a 1:1:1 ratio to one of two dose regimens of pembrolizumab or one regimen of ipilimumab. The group assignment was open-label, and neither investigators nor patients were masked to allocation. The efficacy was analyzed in the intention-to-treat population, while safety was analyzed in all randomly assigned patients who received at least one dose of study treatment. The study’s outcomes revealed that pembrolizumab demonstrated longer overall survival (OS) and PFS than ipilimumab. Also, immune-mediated adverse events were reported, with endocrinopathies being more common in the pembrolizumab group, while colitis was more frequent in the ipilimumab group. Additionally, the study highlighted the need for longer-term follow-up to assess the plateau in the overall survival for pembrolizumab and the outcomes after treatment discontinuation.

Last year, a multicenter study coordinated by the French Society of Pathology involved seven pathology departments in standardizing PD-L1 immunostaining methods and comparing different scoring systems [33]. The aim was to identify a manageable and reliable method for PD-L1 evaluation in metastatic melanoma. Three immunohistochemical platforms were used to assess laboratory-developed tests (QR1, 22C3) and standardized PD-L1 assays [22C3, 28-8, SP142, SP263 (Ventana)] using a training set of tonsil tissue and seven metastatic melanomas in formalin-fixed paraffin-embedded blocks, which showed a range of PD-L1 staining intensities. Additionally, PD-L1 mRNA expression was assessed using an RNAscope and compared with immunohistochemical results in order to offer a reference for PD-L1 expression independent of the immunohistochemical procedure. Seven blinded pathologists graded the PD-L1 immunohistochemical results using the main scoring scales from melanoma clinical trials. Three blinded pathologists validated this procedure on 40 metastatic melanomas (10 primary cutaneous melanomas, 18 lymph node metastases, 11 cutaneous metastases, and 1 lung metastasis). All antibodies/platforms showed high concordances, with the exception of SP142. There were two detected levels of immunostaining intensity: low (28-8, 22C3, and SP142) and high (QR1 and SP263).

For QR1 and SP263, reproducibilities amongst pathologists were higher. QR1, SP263, and 28-8 displayed the highest concordance with mRNA expression. All grading scales showed great concordances between antibodies, and the MEL score showed stronger concordances among pathologists than the other scores [33]. Therefore, to provide a comprehensive evaluation of PD-L1 expression in metastatic melanoma, the authors proposed the assessment of PD-L1 staining to include both PD-L1 expressed by tumor cells (TCs) and combined analyses of TCs and immune cells (ICs) using the MEL score or the CPS. The combined score, in fact, is considered more reproducible, as it is difficult to count TCs exclusively (and not PD-L1-positive macrophages) at the host–tumor interface in the absence of double staining. Additionally, identifying the host–tumor interface in the specimen is important due to the higher density of immune infiltrates and PD-L1-expressing cells than in other tumor areas [33].

In the same year, Yoneta D. et al. [40] analyzed PD-L1 immunoexpression in a series of 56 primary melanomas and 8 paired metastatic lymph nodes from 56 Japanese patients with melanoma (28 acral, 8 mucosal, 18 cutaneous, and 2 unknown). The authors employed the E1L3N, SP142, and 28-8 clones, with a positive threshold of ≥1%. The positive rates for 28-8, E1L3N, and SP142 were 25.0%, 34.0%, and 34.0%, respectively. Positive acral melanoma rates were 10.7% for 28-8, 21.4% for E1L3N, and 21.4% for SP142. The observed positivity rate of mucosal melanoma, in which all three antibodies exhibited reactivity, was found to be 12.5%. Conversely, cutaneous melanomas tested positive for 28-8 in 55.6% of the cases and 66.7% for both E1L3N and SP142. PD-L1-positive tumor cells were significantly correlated with CD4+ TILs and CD8+ TILs. Therefore, staining with E1L3N, SP142, and 28-8 antibodies was within the allowed range, even if the positive rates for E1L3N and P142 were a little bit higher than those for 28-8. CD4+ and CD8+ TILs were shown to be quantitatively associated with PD-L1-positive cancer cells.

Regarding scoring systems, most research evaluating PD-L1 expression in melanoma [38] used the TPS with a 1% cut-off. Nonetheless, several studies have recently adopted the MEL score for evaluating PD-L1 staining employing the 22C3 clone. Therefore, if this clone is available, the MEL score with a >2 cut-off might also be used. Nevertheless, the prognostic and predictive role of PD-L1 expression in CM is really debated, and since the first studies dating back to about ten years ago, its ambivalent role in this pathology has been evident [41,42,43,44].

In 2015, Lipson E.J. et al. [45] and Sunshine J. et al. [46] showed that if 30–40% of unselected melanoma patients who exhibit an objective antitumor response to anti-PD-1/PD-L1 drugs are stratified by PD-L1 expression, those who express PD-L1 tend to have higher average response rates ranging from 50% to 60%, while those who do not express PD-L1 exhibit lower average response rates, typically falling within the range of 10% to 20%. These findings led to FDA approvals for PD-L1 IHC assays, such as the Dako PD-L1 IHC 28-8 PharmDx test as a supplementary diagnosis for patients with metastatic melanoma who may be treated with nivolumab (anti-PD-1). In the same year, Carbognin L. et al. [47] did a meta-analysis investigating the efficacy of anti-PD-1 and anti-PD-L1 treatment in melanoma, lung, and genitourinary cancers. A total of 20 trials were included, of which seven focused on melanoma. The findings of this study indicated a significant increase in overall response rates among tumors with positive PD-L1 expression. The research findings suggested that selecting a PD-L1 positive threshold played a significant role in determining the response to treatment. Specifically, the correlation between tumor response and PD-L1 expression was seen only with a 5% cut-off but not when a cut-off of 1% was used. However, several literature data have reported conflicting findings or showed no significant correlation between PD-L1 expression and clinical outcomes [48,49,50,51,52,53,54,55,56,57,58,59,60,61,62,63,64,65,66,67]. Nonetheless, most of the difficulties in quantifying PD-L1 in melanoma derive from the high melanin content of a significant number of tumor samples and the spatial heterogeneity of its expression [52,64,68].

However, adopting methods to overcome such limitations (such as digital quantification of PD-L1 expression in melanoma tissue specimens) has allowed the demonstration of a trend toward a correlation between PD-L1 expression in melanoma samples and a favorable outcome of PD-1-based immunotherapy [69]. In fact, in 2021, Placke et al. [69] analyzed PD-L1 expression in tissue samples from 156 patients diagnosed with unresectable stage III or IV melanoma prior to PD-1-based immunotherapy and compared digital PD-L1 quantification vs. physician quantification (clone 28-8; cut-off = 5%). The digital method has the advantage that it may be applied independently of the presence of melanin pigmentation. Moreover, PD-L1 expression correlated with therapy outcomes regarding best overall response (BOR), PFS, and OS. Tumor tissue obtained before treatment was collected as part of the multicenter study Tissue Registry in Melanoma (ADOREG/TRIM; CA209-578). Patients with PD-L1-positive tumors showed an improved response compared to negative ones using digital (BOR 50.5% vs. 32.2%; *p* = 0.026) and physician (BOR 54.2% vs. 36.6%; *p* = 0.032) assessment. Tumor PD-L1 expression was associated with an increased PFS and OS by either digital (PFS, 9.9 vs. 4.6 months, *p* = 0.021; OS, not reached vs. 13.0 months, *p* = 0.001) or physician (PFS, 10.6 vs. 5.6 months, *p* = 0.051; OS, not reached vs. 15.6 months, *p* = 0.011) assessment. At a multivariate Cox regression analysis, both digital (PFS, HR = 0.57, *p* = 0.007; OS, HR = 0.44, *p* = 0.001) and physician (OS, HR = 0.54, *p* = 0.016) PD-L1 quantifications were identified as independent predictors of survival upon PD-1-based immunotherapy. The combination of both methods identified a patient subgroup with a particularly favorable therapy outcome (PFS, HR = 0.53, *p* = 0.011; OS, HR = 0.47, *p* = 0.008). Of note, digital PD-L1 quantification was not inferior to the physician’s one, and patients whose tumors were evaluated as PD-L1 positive by both the physicians and the digital algorithm had a BOR on PD-1-based immunotherapy of 60.4%, indicating that tissue PD-L1 expression has predictive value.

Moreover, the same authors [70] have recently aimed to investigate if the type of tumor tissue examined for PD-L1 expression has an impact on the correlation with clinical responses to immunotherapy in melanoma patients. Tumor tissue obtained before treatment was collected as part of the multicenter study Tissue Registry in Melanoma (ADOREG/TRIM; CA209-578) from 448 consecutive patients who underwent PD-1-based immunotherapy for non-resectable stage III or IV metastatic melanoma. The primary study endpoint was BOR, while the secondary endpoints were PFS and OS. All endpoints were correlated with tumor PD-L1 expression (clone 28-8; cut-off ≥ 5%) and stratified based on tissue type. In fact, PD-L1 expression was determined in 95 primary tumors (36.8% positivity), 153 skin/subcutaneous (34.0% positivity), 115 lymph nodes (50.4% positivity), and 85 organ (40.8% positivity) metastases. A statistically significant correlation between tumor PD-L1 expression and BOR was found in lymph node metastases but not in skin/subcutaneous metastases (even if, in this last case, statistical significance was not reached). Moreover, PD-L1 positivity in lymph node metastases was associated with both favorable PFS and OS (both with statistical significance). Of note, PD-L1 positivity in primary tumors only showed a statistically significant correlation with OS. Lastly, PD-L1 expression in skin/subcutaneous metastases showed no relevant correlations with PFS and OS. Therefore, the authors showed that PD-L1 expression in lymph node metastases represents the best therapeutic outcome predictor in non-resectable advanced melanoma, while its assessment on skin/subcutaneous metastases should not be considered for use in therapy decision making.

Therefore, the latest literature data show a tendency towards a correlation between PD-L1 expression in melanoma samples and a positive response to PD-1-based immunotherapy, despite the difficulties related to the interpretation of immunocolorations and the heterogeneity of the expression of this biomarker.

## 5. Biological Basis of Variability in PD-L1 Expressions and Immunotherapeutic Responses to Cutaneous Melanoma

The PD-L1 signal may be received by T cells from several cell types, including APCs and tumor cells. However, the specific cells within the tumor microenvironment that express PD-L1 may have an impact on the biological effects of immune checkpoint inhibition. Several researchers have shown that the correlation between response in melanoma (and other malignancies) and PD-L1 expression is only detected in infiltrating immune cells rather than neoplastic cells [71,72]. A study by Zaretsky J.M. et al. [73] showed that in some individuals who exhibited an initial positive response to anti-PD-1 therapy, melanoma cells demonstrated the presence of PD-L1 before treatment, but upon experiencing a relapse, PD-L1 expression was lost. Conversely, within the same research, it was seen that macrophages and stromal cells expressed PD-L1 during the relapse phase. Hence, although the unique cell type may influence the interaction between PD-1 and PD-L1, the exact correlation between cell type-specific expression of PD-L1 and the response to checkpoint inhibition therapy is not yet fully understood [72]. Moreover, the presence of molecular factors may introduce complexities and hinder a straightforward connection between PD-L1 levels and treatment response. The potential involvement of the immunologic tumor microenvironment in regulating tumor PD-L1 expression is worth considering. The secretion of interferon γ (IFN-γ) by lymphocytes that infiltrate melanoma is linked to the up-regulation of PD-L1 in neoplastic cells [44]. However, melanomas have the ability to develop resistance to IFN-γ signaling, which may diminish the effectiveness of PD-1 or PD-L1 inhibition. Interestingly, Zaretsky J.M. et al. [73] showed that, after a first positive response to anti-PD-1 therapy, loss-of-function mutations in Janus kinase 1 (JAK1) and JAK2 appeared in melanomas that exhibited resistance to further treatment. The absence of functional JAK2 hindered the transmission of signals from IFN-γ, leading to the inability to initiate phosphorylation of signal transducer and activator of transcription 1 (pSTAT1) and PD-L1 expression, ultimately leading to uncontrolled cellular proliferation.

Despite the potential loss of the PD-L1 immune-evasion mechanism in tumor cells, there is a possibility that the efficacy of anti-PD-L1/anti-PD-1 therapy might be maintained via the inhibition of T-cell exhaustion in other locations, such as lymph nodes [72]. This inhibition could potentially enhance the immune response against tumors. One limitation of this paradigm is that tumors lacking PD-L1 expression may have likely developed an alternate way to evade immune detection. Consequently, T cells activated outside the tumor via PD-1/PD-L1 inhibition will need to overcome these additional immune evasion mechanisms when infiltrating the tumor [72].

Based on the aforementioned model and the observed signaling of IFN-γ, it may be inferred that the expression of PD-L1 might potentially serve as a biomarker for various processes occurring inside the tumor [72]. This discovery may provide insight into the common association between tumor PD-L1 expression and immunotherapy response, although not universally consistent.

The gut microbiome represents another possible factor that could predict response to immunotherapy in skin melanoma. In fact, Gopalakrishnan V. et al. in 2018 [74] analyzed the oral and gut microbiota of melanoma patients receiving anti-PD-1 treatment. They observed significant differences in the diversity and composition of the gut microbiome in patients who responded to treatment compared to those who did not. Exploiting metagenomic sequencing, they showed that the gut microbiota in the two groups was functionally different, and the anabolic metabolic pathways were significantly enriched in the patients who responded to treatment. The analysis of immune profiles indicated that patients who exhibited a positive response to treatment had an augmented systemic immune response and increased antitumor immunity. This phenomenon was seen in the case of a favorable gut microbiome and germ-free mice that received fecal transplants from treatment responders. Nonetheless, further research is needed to fully understand the impact of the gut microbiome on the melanoma response to checkpoint blockade immunotherapy.

## 6. Conclusions

A significant amount of scientific research is currently underway with the aim of establishing the most accurate and reliable methodology for assessing the level of PD-L1 immunoexpression in melanoma and its relationship with therapeutic outcomes. Historically, the clinical utility of PD-L1 expression assessment in melanoma has been regarded as controversial, considering the challenges associated with the interpretation of PD-L1 immunocolorations, the variability in the expression patterns of this biomarker and the role of other factors (such as tumor mutational burden, immune cell infiltration, and other immunological checkpoints) in immunotherapy response. However, the most recent literature findings indicate an emerging trend toward a potential association between the expression of PD-L1 in melanoma samples and a favorable response to PD-1-based immunotherapy. In this context, implementing a digital assessment of tumor PD-L1 expression could facilitate diagnostic procedures and enhance the accuracy of treatment outcome prognostication in patients with metastatic melanoma. However, further studies are needed to explore tumor PD-L1 expression using different methods in melanoma patients (for example, employing artificial intelligence techniques to automatically analyze the immunohistochemical expression of PD-L1 in melanoma specimens), and additional comparative molecular analyses of primary tumors and metastases are required to gain a deeper understanding of the tissue-specific mechanisms that contribute to variations in therapy outcomes.

Globally, PD-L1 immunohistochemical determination has strong potential translational applications; further research is needed to dissect the molecular determinants of the expression of such a biomarker, its relationship with tumor microenvironment, and its significance for treatment decisions.

## Figures and Tables

**Figure 1 ijms-25-00676-f001:**
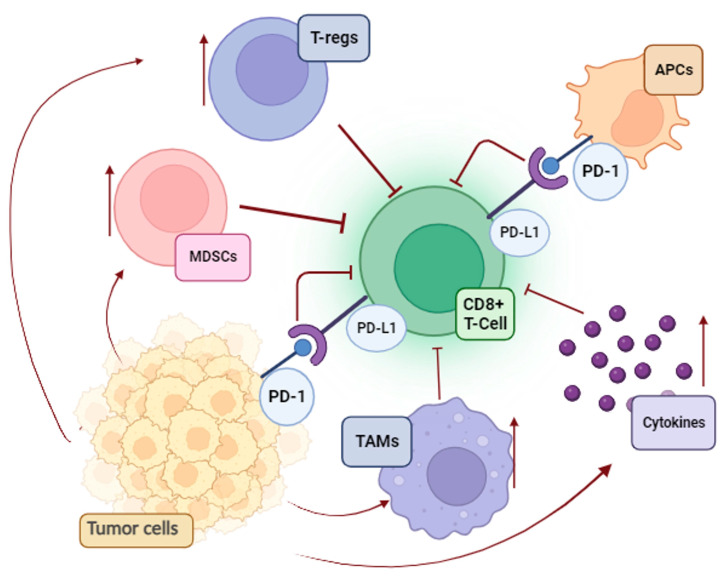
Immunosuppressive pathways in melanoma (MDSCs, myeloid-derived suppressor cells; T-regs, regulatory T cells; APCs, antigen-presenting cells; TAMs, tumor-associated macrophages; PD-L1, programmed death-ligand 1; PD-1, programmed death-1).

**Table 1 ijms-25-00676-t001:** Factors influencing PD-L1 expression in melanoma.

Factors Influencing PD-L1 Expression in Melanoma
Melanoma subtype
Intratumoral heterogeneity
Vertical growth phase
Temporal variability
Prior therapy
Presence of tumor-infiltrating immune cells
Freshly collected vs. archived tumor samples
Inter-laboratory variability
Mutation burden
Immune-related gene expression

**Table 2 ijms-25-00676-t002:** Different PD-L1 scoring systems used in melanoma (TPS, tumor proportion score; CPS, combined positive score; MEL score, melanoma score).

	TPS	CPS	MEL Score	
Definition	$\frac{N.\;\mathrm{of}\;\mathrm{PD}-L1\;\mathrm{positive}\;\mathrm{tumor}\;\mathrm{cells}}{\mathrm{Total}\;n.\;\mathrm{of}\;\mathrm{viable}\;\mathrm{tumor}\;\mathrm{cells}}$	$\frac{N.\;\mathrm{of}\;\mathrm{PD}-L1\;\mathrm{positive}\;\mathrm{tumor}\;\mathrm{and}\;\mathrm{immune}\;\mathrm{cells}}{\mathrm{Total}\;n.\;\mathrm{of}\;\mathrm{viable}\;\mathrm{tumor}\;\mathrm{cells}}$	$\frac{N.\;\mathrm{of}\;\mathrm{PD}-L1\;\mathrm{positive}\;\mathrm{tumor}\;\mathrm{and}\;\mathrm{immune}\;\mathrm{cells}}{\mathrm{Total}\;n.\;\mathrm{of}\;\mathrm{tumor}\;\mathrm{and}\;\mathrm{immune}\;\mathrm{cells}\;\mathrm{close}\;\mathrm{to}\;\mathrm{the}\;\mathrm{neoplasm}}$	
Score	Negative < 1% Positive ≥ 1%	Negative < 1 Positive ≥ 1	Negative Positive	0 1 (0–1%) 2 (≥1% <10%) 3 (≥10% <33%) 4 (≥33% <66%) 5 (≥66%)

## Data Availability

Not applicable.

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
