# Peer review of "Programmed Cell Death Ligand 1 Immunohistochemical Expression and Cutaneous Melanoma: A Controversial Relationship"

_ijms, 2024, doi:10.3390/ijms25010676_

Round 1
Reviewer 1 Report
Comments and Suggestions for Authors
This review paper aims to describe the prognostic and predictive role of PD-L1 expression in CM.
Overall this review paper is not very well structured and organized. The author just summarized different studies in a few sentences, and piled them together into paragraphs.
The two tables in the paper don't really offer any significantly valuable information.
The author should summarize what clinical studies had been conducted related to PD-L1, the number of patients, outcomes, ect.
And when checking PD-L1 expression, what stages of CM or subtypes of CM do different studies use? This information is also worth discussing and comparing.
As the author stated in the conclusion “most useful PD-L1 score ….. Is still a matter of debate”, then the author could summarize the studies that use different scores, including the score, CM type, clinical outcomes, or any detailed information.
Reviewer 2 Report
Comments and Suggestions for Authors
In the manuscript titled "Exploring the Controversial Relationship: Programmed Cell Death Ligand 1 (PD-L1) Immunohistochemical Expression and Cutaneous Melanoma (CM)," the authors aim to delve into the current landscape of knowledge surrounding the prognostic and predictive implications of PD-L1 expression in CM. Additionally, they intend to shed light on potential biological factors contributing to the variability in PD-L1 expressions and their correlation with treatment responses. The paper navigates an intriguing subject matter that holds promise for broader interest. Nevertheless, there are a few issues that merit attention and resolution before the paper can progress towards publication. I have some specific comments that may be useful.
Specific comments:
1. Kindly remove the abbreviation from the title of the paper. A more comprehensive and accessible title will enhance the overall reader experience. Additionally, please ensure that the abstract encapsulates the major outcomes and conclusions derived from the revised material. This should provide a succinct yet informative overview, offering readers a clear glimpse into the key findings and contributions of the manuscript.
2. Figure 1 requires refinement on multiple fronts. Firstly, there is inconsistency in the use of capital letters, with "Tumor cells" capitalized while "cytokines" is not. A uniform style should be applied for clarity. Additionally, the color variations in the depiction of cytokines demand clarification to enhance comprehension. Lastly, the overall quality of the figure appears subpar and would benefit from improvements to better convey the intended information.
3. Kindly refrain from employing abbreviations in titles and subtitles for greater clarity and readability.
4. Many studies are cited, yet they often lack vital information necessary for a thorough assessment of their relevance. It would greatly enhance the comprehensibility of your work if you could incorporate brief discussions on the experimental designs of these studies. Additionally, consider providing details on when these studies were conducted, including the number of participants involved. The inclusion of anthropometric measures, such as age, is imperative, given its unquestionable significance. Age is undoubtedly a crucial factor and warrants a concise discussion within the context of this work.
5. Tables 1 and 2 would benefit from consolidation into a single, more cohesive presentation. Consider merging the data to create a unified table for improved clarity and accessibility. Additionally, the incorporation of a figure could further enhance the visual representation of the information.
6. A thoughtful exploration of limitations is essential for a comprehensive discussion. It would be beneficial to delve into the constraints of the study, acknowledging potential areas of improvement and acknowledging the scope of the research. Additionally, the future perspectives section appears vague; consider providing more specificity and clarity to outline potential avenues for further investigation. Lastly, the conclusion lacks a clear and impactful take-home message.
Round 2
Reviewer 1 Report
Comments and Suggestions for Authors
The paper is much better after revision.